# DIA-Based Quantitative Proteomics in the Flower Buds of Two *Malus sieversii* (Ledeb.) M. Roem Subtypes at Different Overwintering Stages

**DOI:** 10.3390/ijms25052964

**Published:** 2024-03-04

**Authors:** Lijie Li, Xiaochen Lu, Ping Dai, Huaiyu Ma

**Affiliations:** College of Horticulture, Shenyang Agricultural University, Shenyang 110866, China; 2017500060@syau.edu.cn (L.L.); luxiaochen9726@163.com (X.L.); 2019200079@stu.syau.edu.cn (P.D.)

**Keywords:** *Malus sieversii*, flower buds, cold tolerance, DIA-based quantitative proteomics

## Abstract

*Malus sieversii* is considered the ancestor of the modern cultivated apple, with a high value for apple tolerance breeding. Despite studies on the temperature adaptability of *M. sieversii* carried out at a physiological response and the genome level, information on the proteome changes of *M. sieversii* during dormancy is limited, especially about the *M. sieversii* subtypes. In this study, a DIA-based approach was employed to screen and identify differential proteins involved in three overwintering periods of flower buds in two *M. sieversii* subtypes (*Malus sieversii* f. *luteolus*, GL; *Malus sieversii* f. *aromaticus*, HC) with different overwintering adaptabilities. The proteomic analysis revealed that the number of the down-regulated differential expression proteins (DEPs) was obviously higher than that of the up-regulated DEPs in the HC vs. GL groups, especially at the dormancy stage and dormancy-release stage. Through functional classification of those DEPs, the majority of the DEPs in the HC vs. GL groups were associated with protein processing in the endoplasmic reticulum, oxidative phosphorylation, starch and sucrose metabolism and ribosomes. Through WGCNA analysis, tricarboxylic acid cycle and pyruvate metabolism were highly correlated with the overwintering stages; oxidative phosphorylation and starch and sucrose metabolism were highly correlated with the *Malus sieversii* subtypes. This result suggests that the down-regulation of DEPs, which are predominantly enriched in these pathways, could potentially contribute to the lower cold tolerance observed in HC during overwintering stage.

## 1. Introduction

*Malus sieversii* (Ledeb.) M. Roem (*M. sieversii*), also called “Xinjiang wild apple”, is considered the ancestor of the modern cultivated apple (*Malus pumila*), and mainly distributed in the Yili river valley in Xinjiang province, northwest China [1,2]. As a reservoir of genetic diversity, *M. sieversii* is of high economic value for apple tolerance breeding and is usually used as a popular rootstock for its cold-tolerance trait in north-western China [3,4,5,6]. *M. sieversii* in different habitats of Xinjiang has been exposed to cold environments for a long time. Through natural selection and genetic variation, different abilities to tolerate a cold environment have been formed [7]. From the practical point of view, cold tolerance is one of the most important traits when characterizing the germplasm resources or cultivars of deciduous fruit trees. Flower buds of fruit trees are the most vulnerable part during dormancy, thus to determine the cold tolerance of a genotype, the flower buds are worthy to be studied as the weakest link [8]. Fruit trees, which have advanced flower buds formed prior to winter, initiate dormancy by reducing their body water content and respiratory activity in preparation for the colder temperature, while dormancy plays a crucial role in overwintering and productivity [9,10]. The tolerance of apple trees to freezing temperature is variable; the low-temperature tolerance of flower buds changes rapidly in response to both temperature and the stage of wintering dormancy [8,11,12].

Studies on the temperature adaptability of *M. sieversii* have been carried out at the physiological response and the genome level. Zhou et al. (2021) expounded the molecular mechanisms of freezing tolerance after cold acclimation on the basis of genome-wide expression of *M. sieversii* [13]. It has been reported that *M. sieversii* contains many subtypes, which have obvious differences in adaptability to freezing temperatures. Yan et al. (2014) compared and analyzed the adaptability of 28 subspecies of *M. sieversii*; the results showed that *Malus sieversii* f. *luteolus* (GL) had the strongest cold tolerance and *Malus sieversii* f. *aromaticus* (HC) had the weakest cold tolerance among the 28 subtypes [14]. Wang and Qin (2018) reported the physiological response related to cold hardiness of *M. sieversii* in different populations [15]. Further studies proved that the overwintering ability of *M. sieversii* f. *luteolus* was obviously stronger than that of *M. sieversii* f. *aromaticus* at the physiological level [16].

However, information on the proteome changes of *M. sieversii* during dormancy is limited, especially about the *M. sieversii* subtypes. Proteomics is widely regarded as one of the most robust methods, offering an effective approach to explore a systems-based view of how proteins evolve and, consequently, how organisms adapt to diverse abiotic environments [17,18]. Data-independent acquisition (DIA) is a powerful approach for label-free relative protein quantification at the whole proteome level, which is based on the electrostatic field orbital trap, without specifying the target peptide segment, and scanning the number of uniforms; using the spectrum library can achieve qualitative confirmation and quantitative ion screening, and can achieve data backtracking. It has the advantages of high repeatability, high stability, and quantitative accuracy [19,20]. In this study, a comprehensive DIA-based strategy was employed to elucidate an in-depth understanding of the molecular mechanisms of wintering adaptability of flower buds of *M. sieversii*. The overarching aim of this study was to identify the differential proteins involved in three overwintering periods of flower buds in two *M. sieversii* subtypes (*M. sieversii* f. luteolus and *M. sieversii* f. aromaticus), and to implement their functional interpretation. It may be one of the key factors in the difference in cold tolerance between the flower buds of the two *M. sieversii* subtypes. These findings intend to provide a molecular foundation for further research on the mechanism of cold tolerance in *M. sieversii*, and could also be useful for future studies on conservation genetics and potential applications in apple breeding.

## 2. Results

### 2.1. Overview of Proteomic Profiles

To evaluate the cold tolerance of *Malus sieversii* (Ledeb.) M. Roem (*M. sieversii*), the contrasting tolerance ability of flower buds of two *M. sieversii* subtypes were subjected to different overwintering stages. The two subtypes of *M. sieversii* were *Malus sieversii* f. *luteolus* D. F. Cui et L. Wang forma nov. (*M. sieversii* f. *luteolus*; GL) and *Malus sieversii* f. *aromaticus* D. F. Cui et L. Wang forma nov. (*M. sieversii* f. *aromaticus*; HC). The flower buds of GL and HC at different overwintering stages were collected as samples for DIA-based quantitative proteomic analysis (Appendix A). The samples were labeled as GL1 (*M. sieversii* f. *luteolus* at early-dormancy stage), GL2 (*M. sieversii* f. *luteolus* at dormancy stage), GL3 (*M. sieversii* f. *luteolus* at dormancy-release stage), HC1 (*M. sieversii* f. *aromaticus* at early-dormancy stage), HC2 (*M. sieversii* f. *aromaticus* at dormancy stage) and HC3 (*M. sieversii* f. *aromaticus* at dormancy-release stage). In this experiment, a total of 35,114 peptides were identified from apple flower buds that matched with the *Malus* library (Appendix A). Moreover, a total of 6502 protein groups were identified and quantified (Appendix A). The number of peptides and proteins identified in each sample are shown in Appendix A. As shown in Figure 1a, 3715 (56.99%) proteins were identified in both the GL and HC flower buds among three overwintering stages; in addition, there were specific proteins identified in each treatment group, 21, 20 and 132 proteins were specific in GL at early-dormancy stage, dormancy stage and dormancy-release stage, respectively. Similarly, 7, 139 and 94 proteins were specific in HC at those three overwintering stages, respectively. It can be inferred that the difference in cold tolerance between GL and HC was largely related to the types and expression patterns of proteins at the overwintering stages. In a PCA model based on all the samples, the GL and HC samples at the three overwintering stages were clearly separated (Appendix A). It indicated that there were differences between the treatment groups and good reproducibility within the groups. All identified proteins in each treatment groups were quantified and detailed in Appendix A.

### 2.2. Analysis of Differential Expression Proteins (DEPs)

We compared the protein expression of the subtype among three overwintering stages and two subtypes at the same stage, respectively. Proteins with a fold change of above 1.5 or below 0.67 (*p* < 0.05) were considered as differentially expressed proteins (DEPs) in this study. As shown in Figure 1b, there were 348 (161 up-regulated and 187 down-regulated), 1054 (567 up-regulated and 487 down-regulated) and 1494 (762 up-regulated and 732 down-regulated) DEPs identified in GL2 vs. GL1, GL3 vs. GL1 and GL3 vs. GL2; 413 (126 up-regulated and 287 down-regulated), 1079 (418 up-regulated and 661 down-regulated) and 1500 (801 up-regulated and 699 down-regulated) DEPs were identified in HC2 vs. HC1, HC3 vs. HC1 and HC3 vs. HC2, respectively. Moreover, there were 165 (74 up-regulated and 91 down-regulated), 570 (130 up-regulated and 440 down-regulated) and 551 (189 up-regulated and 362 down-regulated) DEPs identified in the groups of HC vs. GL at early-dormancy stage, dormancy stage and dormancy-release stage, respectively, suggesting the different cold responsiveness of these proteins between GL and HC. Detailed information of all the proteins, including protein ID, protein description, protein accessions fold change and *p* values in the *t* tests are provided in Appendix A. Venn diagrams depicted the number of identical or unique DEPs between the different comparison groups, as shown in Appendix A. Volcano pictures also showed significant differences in protein expression between comparison groups; the top 10 with the most significant difference in up-regulated and down-regulated proteins were marked (Appendix A), indicating some crucial proteins or pathways were induced, potentially contributing to the cold tolerance of flower buds.

In order to visually observe the expression patterns of inter-group and intra-group samples, we used the hierarchical clustering algorithm to group and classify the DEPs of the comparison groups, and showed them in the form of heatmaps, as shown in Appendix A. All of the DEPs revealed nine distinct clusters of proteins changing in each treatment group (Figure 2). The results showed that five clusters of protein-expression trends were significantly different between the two *M. sieversii* subtypes, and four clusters (cluster1, 3, 5 and 6) of protein-expression trends in the two subtypes were similar in the three overwintering stages. Among them, cluster 3 had the largest number of DEPs, and 917 proteins had significant changes in abundance; cluster 4 had the smallest number of DEPs, and only 507 proteins had significant changes (detailed information is provided in Appendix A).

### 2.3. Functional Annotation Analysis of Differential Expression Proteins (DEPs)

In order to determine the potential functions of DEPs, the DEPs identified in GL and HC at the three wintering stages were subjected to subcellular localization, domain prediction, gene ontology (GO) and the Kyoto encyclopedia of genes and genomes (KEGG) pathways. Among them, we focused on the function of the DEPs between the two subtypes of *M. sieversii* at the same overwintering stage.

Organelles are important places for proteins to function. In this study, we found that most of DEPs were localized in the cytoplasm, nucleus and chloroplast. And it was worth noting that 22 proteins were in the mitochondria at the early-dormancy stage (HC1 vs. GL1); however, the number of them significantly increased at dormancy stage (HC2 vs. GL2) and dormancy-release stage (HC3 vs. GL3), indicating that the changes of proteins localized in the mitochondria were closely related to the cold resistance of flower buds in the two *M. sieversii* subtypes (detailed information of subcellular localization is provided in Appendix A including all the comparison groups). Protein domains represent that proteins have different biological functions, which is of great significance to studying the key functional regions of proteins and their potential biological roles [21]. In this study, the protein domain was predicted and the results are shown in Appendix A.

Functional classification of the identified DEPs was performed by GO analysis including the biological process (BP), cellular component (CC) and molecular function (MF). As shown in Figure 3, GO analysis showed that the top two processes of the BP, MF and CC terms were consistent among the three comparison groups. Namely, metabolic process and cellular process were the major BP terms; catalytic activity and binding were the dominant MF terms; and cell, cell part, organelle, membrane, membrane part and protein-containing complex were the major CC terms. The number of DEPs in the top two processes of BP and CC terms gradually increased with the development of the dormancy stage, while the number of DEPs in the top two process of the MF term was the highest at the dormancy stage. The comparison of the subtype at three overwintering stages was consistent with the results of the three processes described above (Appendix A). In addition, we also performed GO enrichment analysis for DEPs. It was noted that ribosomes, the structural constituents of ribosomes and translation were significantly enriched in cellular components, molecular functions and biological processes, respectively, at the dormancy-release stage between GL and HC.

To understand the important metabolic pathways in the two subtypes at three overwintering stages, KEGG enrichment analysis on cold responsive proteins from GL and HC were performed. The results indicated that protein processing in the endoplasmic reticulum and oxidative phosphorylation at the early-dormancy stage (Figure 4a) and at the dormancy stage (Figure 4b) and ribosome and starch and sucrose metabolism at the dormancy-release stage (Figure 4c) were considerably enriched. In order to further understand the types of pathways of DEP enrichment, we classified the DEPs. The results showed that protein processing in the endoplasmic reticulum, oxidative phosphorylation, ribosome and starch and sucrose metabolism belonged to folding, sorting and degradation pathways, energy metabolism, translation and carbohydrate metabolism in Level 2, respectively. Interesting, we also found that the changes of the number of enriched proteins in protein processing in the endoplasmic reticulum, oxidative phosphorylation and starch and sucrose metabolism were similar. The numbers of enriched proteins were the highest at the early-dormancy stage and dormancy stage, and then decreased at the dormancy-release stage. However, the number of enriched proteins in the ribosome continued to increase, with 65 proteins at the dormancy-release stage. These results suggested that the difference in protein expression between the two subtypes caused changes in metabolic pathways to cope with cold stress, which should be one of the main reasons for the difference in cold tolerance between GL and HC. In addition, the physiological stage also had a certain impact on it. We observed significant enrichment pathways of the same subtype at the three wintering stages, including ribosomes (the number of DEPs was the most), biosynthesis of cofactors and carbohydrate metabolism (including phenylpropanoid biosynthesis and the citrate cycle (TCA cycle), etc.) (Appendix A).

### 2.4. Weighted Gene Co-Expression Network Analysis (WGCNA)

To further understand the DEPs in the flower buds of GL and HC at the three overwintering stages, and to determine the key proteins that affected the cold tolerance of the flower buds, we analyzed the DEPs using weighted gene co-expression network analysis (WGCNA) [22,23]. In this study, all the DEPs based on expression patterns were divided into different expression modules as shown in Figure 5a; 10 modules (represented in 10 colors) were identified (detailed information is provided in Appendix A including all the comparison groups), and the turquoise module contained the largest number of proteins, with 1620 proteins. In addition, we also conducted correlation analysis for each module and trait, and the results are shown in Figure 5b. Based on the correlation coefficient, it could be inferred that the turquoise module was highly correlated with the overwintering stages, and the yellow module was highly correlated with the subtypes.

All proteins of the turquoise module (Figure 5c) or yellow module (Figure 5d) were analyzed for enrichment of the KEGG pathway. The results showed that ribosome was the most main enriched pathway, whether it was in the turquoise or yellow module. In addition, we also found the citrate cycle (TCA cycle) and pyruvate metabolism were the main enriched pathways in the turquoise module, while oxidative phosphorylation and starch and sucrose metabolism were the main enriched pathways in the yellow module. This result was consistent with those mentioned in the previous study, and further illustrated that changes in the proteins of these pathways may be the main reasons for affecting the cold tolerance of flower buds in both subtypes at three overwintering stages.

## 3. Discussion

Plants have developed several adaptability mechanisms to tolerate wintering environments. Physiological measurements have confirmed that the subtypes of M. sieversii present have obvious differences in adaptability of cold temperature [14,15,16]. In order to understand the molecular information on specific proteins and associated biological pathways that contribute to cold tolerance and susceptibilities, the proteomic profiles of the flower buds of *M. sieversii* f. *luteolus* (GL, with high cold tolerance) and *M. sieversii* f. *aromaticus* (HC, with low cold tolerance) in three overwintering stages were examined by a DIA-based approach. The three overwintering stages were early-dormancy, dormancy and dormancy-release. The dormancy-release stage means that the flower buds have completed natural dormancy and are in a forced dormancy state. In the present experiment, both in the GL and HC, the number of DEPs at the dormancy-release stage was the highest, and the number of up-regulated DEPs in GL was higher than that of the down-regulated DEPs, while the opposite was true for HC. In the three HC vs. GL groups, the number of the down-regulated DEPs was clearly higher than that of the up-regulated DEPs, especially at the dormancy stage. This reflects that HC is more inhibited by low temperature than GL at the protein level. In the following discussion, we discuss the main changing proteins and underlying/potential biological processes that were affected by the genotype and the low temperature.

### 3.1. DEPs Involved in Protein Synthesis

The balance between protein synthesis and degradation is the key to maintaining metabolism balance in plant cells, however, this balance is often disrupted by adverse environmental stress [24]. Ribosomal proteins are considered to be important components of the stress response in plants, and overexpression of ribosomal proteins enhances low temperature tolerance in winter rye leaves [25]. Shan et al. (2018) found that the expression of 6 ribosomal proteins was inhibited, while the expression of 10 proteins was increased by long-term drought induction [26]. In this study, it was observed that more DEPs were enriched in the ribosome, which were not only related to the subtypes of *M. sieversii*, but also to them at the three overwintering stages. As shown in Figure 6, five DEPs of ribosomal proteins were significantly down-regulated in HC1 vs. GL1 (at the early-dormancy stage). The expression of all the DEPs in the ribosome decreased significantly at the dormancy and the dormancy-release stages, except for A0A498ICY4 and A0A498I572. These results were similar to Xu et al. (2018), which showed that the expressions of ribosomal proteins had significant differences between two varieties of *Brassica rapa* L. with different cold tolerance [27]. Some experiments have proven that the up-regulation of ribosomal proteins help plants to resist adverse stress [28,29]. Therefore, we speculated that the DEPs in the ribosome may be one of the reasons why HC has lower cold tolerance than GL. In addition, we also found that the number of ribosomal DEPs increased significantly at the dormancy (HC2 vs. GL2) and the dormancy-release (HC3 vs. GL3) stages, which were 16 and 65, respectively, compared with the early-dormancy stage. At the same time, the same changing trend was observed in the comparison group of the same subtype at the three overwintering stages. These results suggest that both cold-tolerant and cold-sensitive subtypes of *M. sieversii* need to synthesize more proteins to cope with low temperature during forced dormancy.

### 3.2. DEPs Involved in Carbohydrate and Energy Metabolism

Carbohydrate metabolism, including the glycolytic (EMP), tricarboxylic acid cycle (TCAC) and pentose phosphate pathway (PPP) are crucial sources of energy supply and have been proven to be important survival strategies for plants in response to various stresses [30,31]. It has been found that the EMP and TCAC pathways can provide necessary energy for various life activities in plant cells, and they are also affected by a variety of adverse environment [32,33]. In this study, the DEPs which are involved in EMP, TCAC and PPP were summarized, as shown in Table 1. Obviously, the group of HC2 vs. GL2 had the highest number of DEPs (at the dormancy stage). All the DEPs involved in EMP, TCAC and PPP at the dormancy stage were down-regulated in HC2 vs. GL2, except for aldehyde dehydrogenase (Q9ZPB7), which may have been one of the reasons why HC was less tolerant to low temperature than GL.

In the EMP, hexoses are oxidized to generate ATP, reductant and pyruvate; hexoses produces building blocks for anabolism [34]. Previous reports have confirmed that fructose-6-phosphate kinase and pyruvate kinase play important roles in EMP [35]. It has been found that low-temperature stress causes the activity decreases of pyruvate kinase, hexokinase and glyceraldehyde-3-phosphate dehydrogenase in corn embryos [36]. In this study, it was observed that the expressions of glyceraldehyde-3-phosphate dehydrogenase (A0A498JQC8 and A0A540KTG1) and phosphopyruvate hydratase (A0A498KN40 in HC3 vs. GL3) were significantly up-regulated in HC compared to GL. Since these two enzymes (glyceraldehyde-3-phosphate dehydrogenase and phosphopyruvate dydratase) catalyze a bidirectional reaction, the increased protein expression of these two enzymes does not imply increased activity of the EMP pathway. Pyruvate kinase adjusts the carbohydrate flux in glycolysis [37]. The expression of pyruvate kinase (A0A540NAK7) in HC2 vs. GL2 was down-regulated, which means the inhibition on pyruvate production due to cold stress. However, the change of pyruvate kinase expression was complicated in the HC3 vs. GL3 group, the expressions of A0A498IKR9 and A0A498K862 were un-regulated, and the expression of A0A498JQT3 was down-regulated. And the expression of pyruvate decarboxylase was up-regulated markedly in the HC3 vs. GL3 group. A combination of the expression changes of pyruvate kinase and pyruvate decarboxylase, even with the increase in pyruvate production, meant the conversion of pyruvate to acetaldehyde was increased (catalyzed by pyruvate decarboxylase) at the dormancy-release stage. Wang et al. (2021) proposed that the expression of genes involved in the EMP and TCA pathways can influence the anaerobic and aerobic respiratory efficiency of *Vitis amurensis*, thereby facilitating a reduction in sugar consumption in dormant buds and enhancing cold tolerance during winter [38]. This study provides a valuable reference for understanding the mechanism by which GL flower buds regulate cold tolerance.

The pyruvate produced through glycolysis is transferred into the TCAC, then transported down the electron transfer chain and subsequently returned to the mitochondrial matrix by ATP synthase [39,40]. Malate dehydrogenase, succinate dehydrogenase and citrate synthase are the crucial enzymes involved in the TCAC, and the pyruvate dehydrogenase complex, as a multi-enzyme complex in the mitochondrial matrix, plays a key role in the energy metabolism of the mitochondrial respiratory chain [41]. In this study, we observed an increase in the expression of malate dehydrogenase (A0A498I8M1 in HC1 vs. GL1 and HC3 vs. GL3) and pyruvate dehydrogenase (A0A498IVM7 in HC3 vs. GL3), and a decrease in the expression of succinate dehydrogenase (A0A498KGN9 and A0A498HDN1) citrate synthase (A0A540NBC6 and A0A498HE04) in HC2 vs. GL2. This finding suggests that TCAC activity is significantly reduced under cold stress in HC at the dormancy stage, and then impacts ATP production in the mitochondria. Previous studies have shown that stress reduced TCAC activity, which led to the reduction of TCAC intermediates [42,43]. As is known to all, TCAC is a metabolic hub necessary for ATP production, and its intermediates are necessary for providing precursors used in many biosynthetic pathways involved in carbohydrates, fatty acids and amino acids metabolism [44,45]. Therefore, we speculated that the significant down-regulation of proteins involved in TCAC in HC is one of the main reasons for the low cold-tolerance of HC during overwintering, compared to GL.

PPP is also an important channel for providing energy and precursor substances for plants [46,47]. In this study, important proteins related to PPP, including glucose-6-phosphate 1-dehydrogenase (A0A498ITD4, A0A498HRZ9 and A0A498J4J5), phosphoenolpyruvate carboxykinase (A0A1B1UZZ5) and ATP-dependent-6-phosphofructokinase (A0A498KQA5 and A0A540M0H0), were down-regulated markedly in the HC2 vs. GL2 group, which was similar to that observed by Wang et al. (2021) in the response mechanism of pepper seedlings to heat stress [46], indicating the flower buds of HC suffered severe oxidative damage at the dormancy stage. However, the expressions of phosphopyruvate hydratase (A0A498KN40) and pyrophosphate--fructose 6-phosphate 1-phosphotransferase (A0A498JT62) in the HC3 vs. GL3 group were significantly up-regulated, inferring that cold-sensitive flower buds may actively respond to cold stress by increasing PPP activity compared with cold-tolerant flower buds at the dormancy-release stage.

Starch and sucrose metabolism plays an important role in the cold tolerance of plants. Among them, sugar is a signaling molecule involved in a variety of physiological processes such as hormone synthesis, plant growth and development and the stress response, and as a nutrient providing energy and a protective agent against cold damage [48,49]. In this study, the DEPs involved in starch and sucrose metabolism were summarized, as shown in Table 2. Obviously, the groups of HC2 vs. GL2 and HC3 vs. GL3 had a higher number of DEPs (at the dormancy stage and dormancy-release stages) than that in the HC1 vs. GL1 group (at the early-dormancy stage), suggesting that the flower buds of *M. sieversii* may require more proteins involved in starch and sucrose metabolism to cope with low-temperature stress at the dormancy stage and dormancy-release stage. It has been reported that sucrose synthase catalyzes the reversible production of sucrose into uridine diphosphate glucose and fructose, and that sucrose phosphate synthase is considered as a crucial enzyme in sucrose synthesis [50]. In the present study, we observed that proteins related to starch and sucrose metabolism, including starch synthase (B2LUN5), sucrose synthase (A0A498HVP0 and A0A498KKD2), glucose-1-phosphate adenylyltransferase (A0A498JXG5 and A0A498J684) and alpha-1,4 glucan phosphorylase (A0A498KD22 and A0A498J394), were up-regulated in the HC vs. GL group at the dormancy-release stage, indicating that starch and sucrose metabolism in HC flower buds was more sensitive than that in GL in responding to low-temperature stress.

Oxidative phosphorylation takes place in the mitochondria, provides adenosine triphosphate (ATP) for driving cellular function in plants and is able to regulate intracellular oxygen balance to resist oxidative stress [51,52]. The initial phase involves the transfer of electrons from reducing equivalents to molecular oxygen. This mechanism relies on the respiratory electron transfer chain situated in the inner mitochondrial membrane. The transfer of electrons is linked to the creation of an electrochemical proton gradient across the membrane. Subsequently, this proton gradient is harnessed by the ATP synthase complex in the second phase to phosphorylate adenosine diphosphate (ADP). The produced ATP can then be transported out of the mitochondria and distributed throughout the entire eukaryotic cell [53]. In this study, we found that oxidative phosphorylation was one of the major enrichment pathways of DEPs, indicating that the changes of DEPs in oxidative phosphorylation may be one of the main reasons for the different cold tolerance of the two *M. sieversii* subtypes flower buds at the three overwintering stages. In order to clearly observe the fold-changes of DEPs in oxidative phosphorylation at the three overwintering stages between HC and GL (HC1 vs. GL1, HC2 vs. GL2 and HC3 vs. GL3), we summarized them in Table 3. It was observed that the group of HC2 vs. GL2 had the highest number of DEPs (at the dormancy stage), and all the DEPs in the HC2 vs. GL2 group involved in oxidative phosphorylation were significantly down-regulated. Fei et al. (2021) found that the changes of ATP synthase subunits might influence the metabolic rates of ATP synthase and affect energy output, and then induce potential changes, so the up-regulated ATP synthase might lead to enhance the cold resistance of *K. obovate* [54]. In the present study, the expressions of ATP synthase (A0A0U2N8T4 and A0A1C8YB78 in HC2 vs. GL2, A0A498KNH1 in HC3 vs. GL3) were down-regulated markedly, therefore, we speculated that GL has a higher cold tolerance than HC due to higher protein expression in oxidative phosphorylation at the dormancy stage. NADH dehydrogenase complex involves in the electron transport chain during cellular respiration, which is responsible for oxidizing NADH to NAD^+^ while transferring electrons to electron acceptors downstream [53,55]. NADH dehydrogenase complex is composed of more than 41 subunits [56]. In this study, all the NADH dehydrogenase complex subunits were significantly down-regulated in the HC2 vs. HC2 and HC3 vs. GL3 groups, except for NADH dehydrogenase (ubiquinone) 1 beta subcom-plex subunit 10 (A0A498HR73 and A0A540L321) in the HC1 vs. GL1 group. Previous research has found that the NADH dehydrogenase complex affects the low-temperature sensitivity in rice [57,58]. Therefore, we inferred that this may be one of main reasons why HC has a lower low-temperature tolerance than GL.

### 3.3. DEPs Involved in Other Metabolism

In this study, phenylpropanoid biosynthesis and flavonoid biosynthesis in the flower buds of HC and GL changed to different degrees at the three overwintering stages, as shown in Table 4. Peroxidase, a well-known antioxidant enzyme, plays a crucial role in the cellular detoxification of H_2_O_2_. Not only does it efficiently shield cellular components such as proteins and lipids from oxidation, but it is also indispensable for various crucial cellular functions such as lignification, suberization, cell elongation, growth, regulation of cell wall biosynthesis and plasticity [59,60]. In our study, the fold-changes of DEPs for peroxidase in the HC vs. GL group was complicated at the three overwintering stages. The expressions of peroxidase (A0A498KCP1, A0A498I2P0 and A0A498IIU4) were down-regulated in the HC1 vs. GL1 and HC2 vs. GL2 groups; however, there still existed some peroxidases that were up-regulated, A0A498KKN7 in the HC2 vs. GL2 and HC3 vs. GL3 groups and A0A498KLW3 in the HC3 vs. GL3 group. This may be related to the forms of peroxidase according to Mathai et al. (2020) [61].

## 4. Materials and Methods

### 4.1. Plant Materials

*Malus sieversii* f. *luteolus* D. F. Cui et L. Wang forma nov. (GL) and *Malus sieversii* f. *aromaticus* D. F. Cui et L. Wang forma nov. (HC) were cultivated by the resource nursery of Kazakh autonomous prefecture agricultural science research Institute (81°23 E, 43°55′ N), Yining, Xinjiang, China. The flower buds were collected at three overwintering periods, with samples taken on 10 October 2020 (pre-dormancy), 25 December 2020 (dormancy) and 1 April 2021 (dormancy end). Sampling was conducted in triplicate for each period for the two *M. sieversii* subtypes; the samples were labeled as GL1, GL2, GL3, HC1, HC2 and HC3, respectively. All of the collected samples were frozen immediately in liquid nitrogen and stored at −80 °C, and used for DIA-based quantitative proteomic analysis.

### 4.2. Protein Preparation and Digestion

Flower buds (approximately 3.0 g) of each sample was ground into powder with liquid nitrogen. Then, SDT buffer (4% SDS, 100 mM DTT, 150 mM Tris-HCl pH 8.0) was added to the samples directly and the sample lysate was further sonicated. After being centrifuged at 14,000× *g* for 40 min, the supernatant was quantified with the BCA Protein Assay Kit (Bio-Rad Laboratories, Hercules, CA, USA). The sample was stored at −80 °C. An equal aliquot from each sample in this experiment was pooled into one sample in this experiment for DDA library generation and quality control.

### 4.3. Filter-Aided Sample Preparation (FASP Digestion) Procedure

As per the protocol, 200 μg of proteins from each sample were dissolved in SDT buffer (4% SDS, 100 mM DTT, 150 mM Tris-HCl pH 8.0). UA buffer (8 M Urea, 150 mM Tris-HCl pH 8.0) was utilized in multiple rounds of ultrafiltration (Microcon units, 10 kD) to discard detergent, DTT and other low-molecular-weight components by repetition. Next, an equal volume of UA buffer containing100 mM IAA and 100 μL iodoacetamide was added to block reduced cysteine residues; the mixture was incubated in darkness for 30 min. The filters were rinsed thrice with UA buffer and twice with 100 μL 25 mM NH_4_HCO_3_ buffer. Subsequently, the protein suspensions were enzymatically degraded with 4 μg of trypsin (Promega Biotech Co., Ltd., Beijing, China) in 40 μL 25 mM NH_4_HCO_3_ buffer overnight at 37 °C. Enzymatic reaction products, i.e., peptides, were collected through filtration. The peptide concentration was measured using a specific extinction coefficient of 1.1 (g·L^−1^) at a wavelength of 280 nm, established via spectrometric analysis of tryptophan and tyrosine content in vertebrate proteins. Digested peptides were divided into ten fractions using the High pH Reversed-Phase Peptide Fractionation Kit, Thermo Scientific™ Pierce™ (Thermo Scientific, Madison, WI, USA). Each fraction was concentrated under vacuum and restored in 0.1% (*v*/*v*) formic acid. Desalted peptides were concentrated on C18 Cartridges, Empore™ SPE Cartridges C18 (standard density), bed I.D. 7 mm, volume 3 mL, Sigma, Aldrich chemie GmbH, Germany) then restored in 0.1% (*v*/*v*) formic acid. iRT-Kits (Biognosys, Cambridge, MA, USA) were applied to account for relative retention time variations.

### 4.4. Data Dependent Acquisition (DDA) Mass Spectrometry Assay

For the DDA library construction, all fractions were analyzed using an interfaced Thermo Scientific Q Exactive HF X mass spectrometer and Easy nLC 1200 chromatography system (Thermo Scientific, Madison, WI, USA). The peptide (1.5 μg) was loaded onto an EASY-SprayTM C18 Trap column (P/N 164946, 3 μm, 75 μm × 2 cm, Thermo Scientific, Madison, WI, USA), and paired with the EASY-SprayTM C18 LC Analytical Column (ES802, 2 μm, 75 μm × 25 cm, Thermo Scientific, Madison, WI, USA) for chromatographic separation. A linear gradient of buffer B (80% acetonitrile and 0.1% Formic acid) at a flow rate of 250 nL/min maintained for 90 min ensured effective separation. Positive ionization, with scanning ranging from 300 to 1800 *m*/*z*, enabled detection of molecular weight. MS1 resolution was 60,000 at 200 *m*/*z*, with a target of AGC (automatic gain control) set at 3e6; the maximum IT was 25 ms, and dynamic exclusion was 30.0 s. Each full MS–SIM scan was preceded by 20 dd MS2 scans, with MS2 resolution at 15,000; the AGC target was 5e4, maximum IT was 25 ms and normalized collision energy was 30 eV.

### 4.5. Data Independent Acquisition (DIA) Mass Spectrometry Assay

Each sample’s peptides was evaluated through LC-MS/MS functioning in the data-independent acquisition (DIA) mode by Shanghai Applied Protein Technology Co., Ltd. Each DIA cycle incorporated one full MS–SIM scan, while the 30-scan involvement spanned a range of 350–1800 *m*/*z*. The settings included the following: the SIM full scan resolution was 120,000 at 200 *m*/*z*, AGC 3e6, maximum IT 50 ms, profile mode; DIA scans had a resolution of 15,000, AGC target 3e6, Max IT auto, normalized collision energy set at 30 eV. Running time was 90 min with a linear gradient of buffer B (80% acetonitrile and 0.1% Formic acid) at a flow rate of 250 nL/min. To monitor MS performance, QC samples (a pooled sample from an equal aliquot of each sample in the experiment) were injected into DIA mode at the beginning of the MS study and after every 6 injections throughout the experiment.

### 4.6. Mass Spectrometry Data Analysis

The DIA data was processed with the Spectronaut Pulsar XTM (version 12.0.20491.4), searching the above constructed spectral library. The main software parameters were set as follows: retention time prediction type was set as dynamic iRT, interference on MS2 level correction was enabled and cross run normalization was also enabled. All the obtained results were subsequently filtered based on a Q value cutoff of 0.01 (equivalent to FDR < 1%).

### 4.7. Bioinformatic Analysis

The cluster 3.0 (http://bonsai.hgc.jp/~mdehoon/software/cluster/software.htm (accessed on 20 June 2021)) and Java Treeview (http://jtreeview.sourceforge.net (accessed on 21 June 2021)) were used to performing hierarchical clustering analysis. The fuzzy c-means (FCM) algorithm of Mfuzz software (the R (Version 3.4) Mfuzz package) was used for analysis, which was divided into different expression modules according to the expression trend of all the proteins. CELLO (http://cello.life.nctu.edu.tw/ (accessed on 21 June 2021)) was used to predict protein subcellular localization. Protein sequences were searched using the InterProScan software (version 90.0) to identify protein domain signatures from the InterPro member database Pfam [62]. The GO terms and KEGG pathways were annotated by Blast2GO (Version 2.5.0) and Kyoto Encyclopedia of Genes and Genomes (KEGG) database (http://geneontology.org/ (accessed on 22 June 2021)), respectively. Enrichment analyses were applied based on the Fisher’ exact test, and only functional categories and pathways with *p*-values under a threshold of 0.05 were considered as significant.

WGCNA algorithm (weighted gene co-expression network analysis) is a common algorithm for constructing weighted co-expression networks. First, assume that there is a protein and scale-free distribution network, then define the protein expression correlation matrix and the protein network formed by the adjacency function, and then calculate the otherness coefficient of different nodes. On the basis of constructing a hierarchical clustering tree (hierarchical clustering tree), different branches of the clustering tree represent different protein modules (module); if the module protein expression degree is high, but belongs to different modules of the protein, there is a low degree of expression. Finally, the correlation between protein modules and specific phenotypes or diseases is explored to finally identify the target proteins and protein networks for disease treatment. This experiment uses the R package WGCNA (R Version 3.4) to write a script to build a weighted co-expression network [22,23].

## 5. Conclusions

In this study, we reported the protein profiles of two *M. sieversii* subtypes (HC, with low cold tolerance; GL, with high cold tolerance) at three overwintering stages (early-dormancy, dormancy and dormancy-release) through DIA quantitative proteomics identification technology. A total of 35,114 peptides and 6502 proteins were identified. The study found that the number of specific proteins in HC was the highest at the dormancy stage; however, the number of specific proteins in GL increased with the development of the dormancy stage. At the same time, we compared the protein expression of the subtype among three overwintering stages and two subtypes at the same stage, respectively. The functional annotation of differential expression proteins (DEPs) was mainly involved in pathways related to protein processing in the endoplasmic reticulum, ribosome, carbohydrate metabolism, oxidative phosphorylation, starch and sucrose metabolism. The number of the down-regulated DEPs was obviously higher than that of the up-regulated DEPs in the HC vs. GL groups involved in those pathways, especially at the dormancy stage and dormancy-release stage. Most of down-regulating DEPs were enriched in the ribosome, carbohydrate metabolism (glycolytic, tricarboxylic acid cycle, pentose phosphate pathway and starch and sucrose metabolism) and energy metabolism (oxidative phosphorylation) in the HC vs. GL groups; this may be the reason for the low cold tolerance of HC during overwintering.

## Figures and Tables

**Figure 1 ijms-25-02964-f001:**
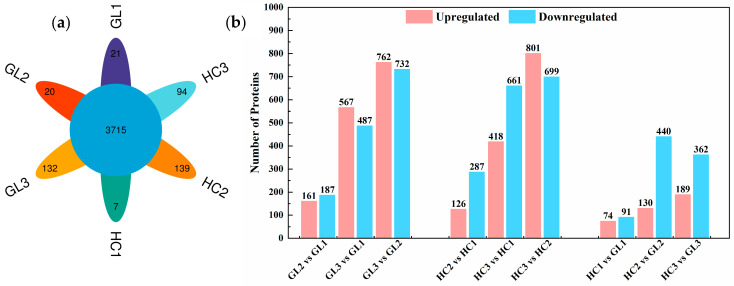
(**a**) Venn diagram of identified proteins in *M. sieversii* f. luteolus (GL) and *M. sieversii* f. aromaticus (HC) flower buds at three overwintering stages. (**b**) Comparison of the number of up-regulated and down-regulated proteins based on the two subtypes and the overwintering stages. GL1, *M. sieversii* f. luteolus at early-dormancy stage; GL2, *M. sieversii* f. luteolus at dormancy stage; GL3, *M. sieversii* f. luteolus at dormancy-release stage; HC1, *M. sieversii* f. aromaticus at early-dormancy stage; HC2, *M. sieversii* f. aromaticus at dormancy stage; HC3, *M. sieversii* f. aromaticus at dormancy-release stage. The same below.

**Figure 2 ijms-25-02964-f002:**
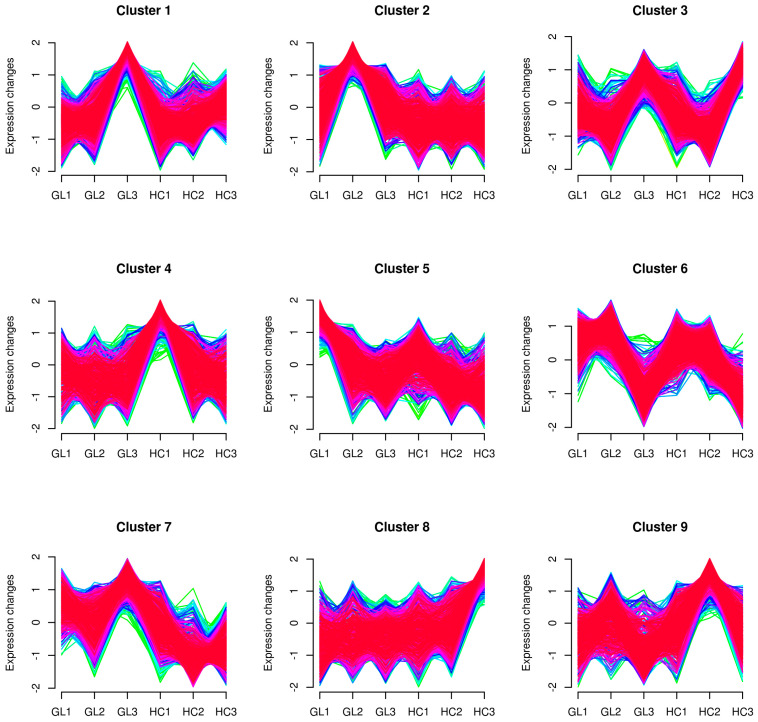
Clustering of all identified proteins according to fuzzy c-means (FCM) analysis of GL and HC flower buds at three overwintering stages. The *X*-axis and *Y*-axis correspond to the group, and expression levels after homogenization, respectively. The lines of each cluster represent a class of proteins with consistent expression trends.

**Figure 3 ijms-25-02964-f003:**
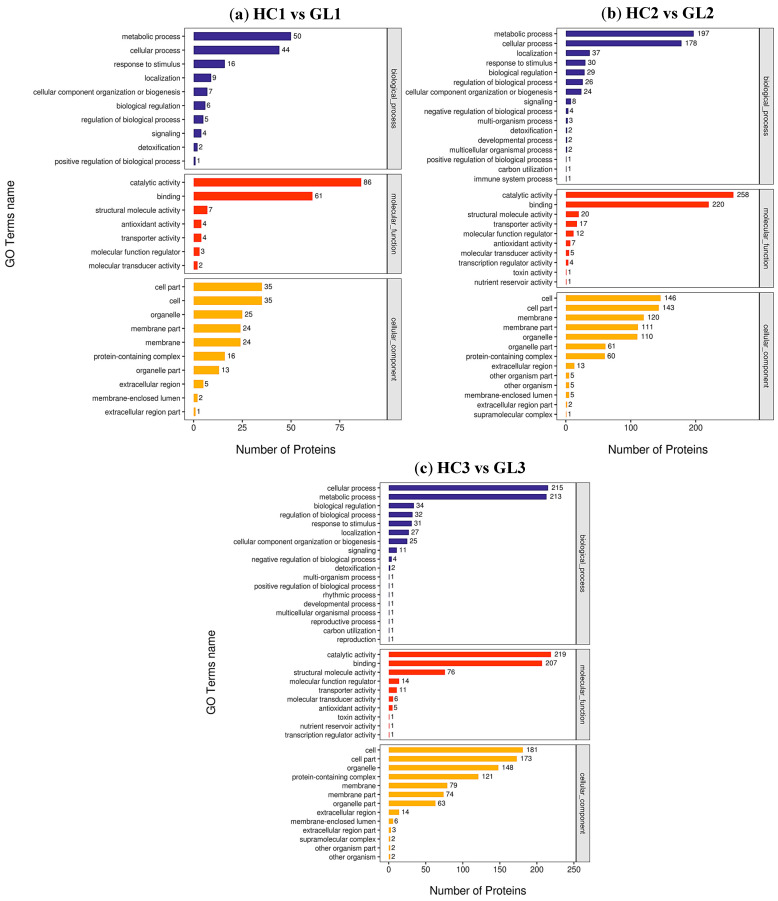
GO analysis of DEPs at three wintering stages between two cultivars. (**a**) GO classification of DEPs at early-dormancy stage between HC and GL; (**b**) GO classification of DEPs at dormancy stage between HC and GL; (**c**) GO classification of DEPs at dormancy-release stage between HC and GL. The abscissa and ordinate correspond to the number of DEPs and the name of GO terms.

**Figure 4 ijms-25-02964-f004:**
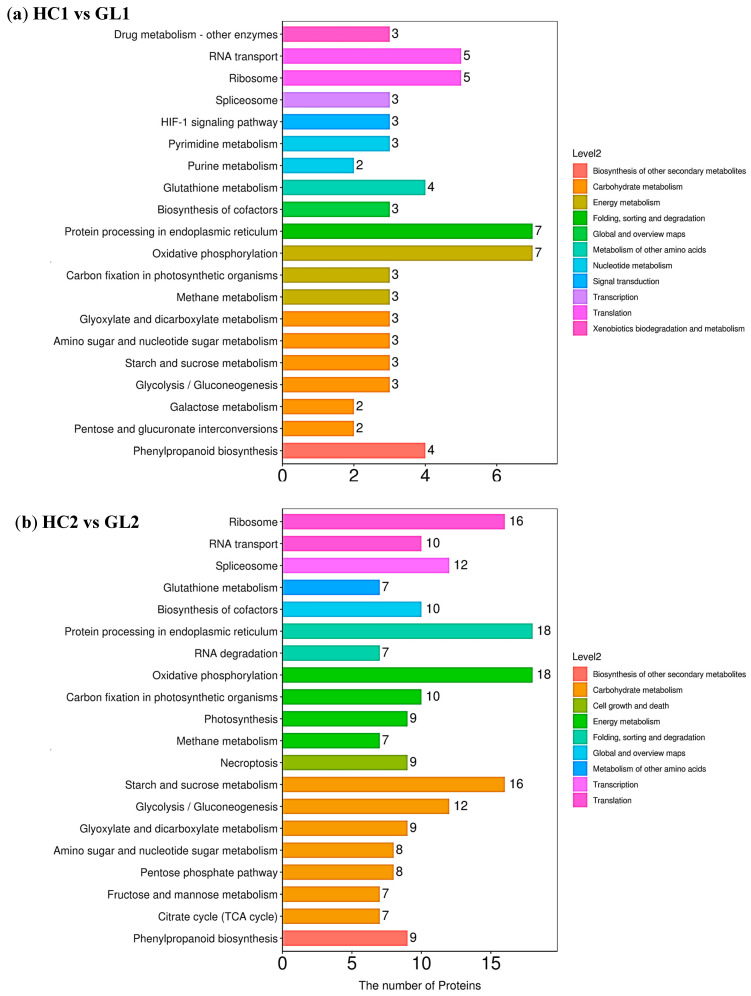
Annotation and attribution of the KEGG pathway for DEPs at three overwintering stages between the two sybtypes. (**a**) KEGG pathway classification of DEPs at early-dormancy stage between HC and GL; (**b**) KEGG pathway classification of DEPs at dormancy stage between HC and GL; (**c**) KEGG pathway classification of DEPs at dormancy-release stage between HC and GL. The abscissa and ordinate correspond to the number of DEPs and the name of KEGG pathway.

**Figure 5 ijms-25-02964-f005:**
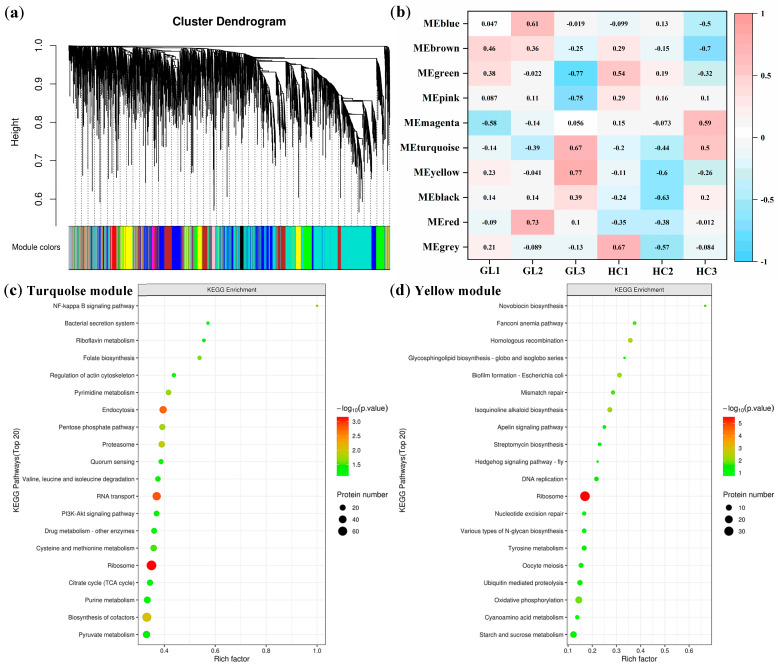
Weighted gene co-expression network analysis: (**a**) Hierarchical clustering dendrogram shows co-expression modules that are color-coded. Module colors represent the final modules. Each branch in the hierarchical tree or each vertical line in color bars corresponds to a single protein. (**b**) A module-trait correlation plot displays the correlation between each module and each trait attribute. Negative correlations are represented in blue, while positive correlations are shown in red. (**c**) KEGG enrichment analysis of the proteins in the turquoise module. (**d**) KEGG enrichment analysis of the proteins in the yellow module.

**Figure 6 ijms-25-02964-f006:**
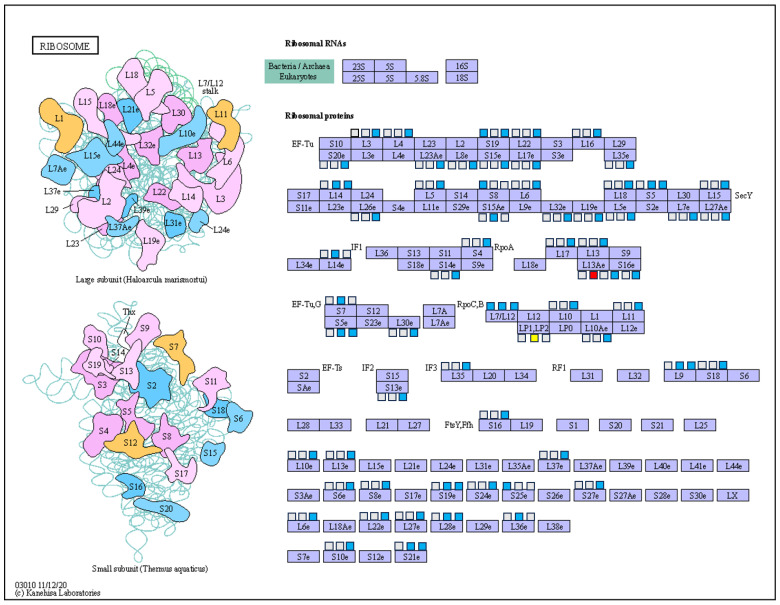
Map of a ribosome. The left square (□) displays HC1 vs. GL1, the middle square (□) displays HC2 vs. GL2 and the right square (□) displays HC3 vs. GL3; red color and blue color represent differentially up-regulated and down-regulated proteins, respectively. Yellow represents DEPs including up-regulated and down-regulated proteins. Grey represent no significant difference in protein expression.

**Table 1 ijms-25-02964-t001:** List of the fold-changes of DEPs in EMP, TCAC and PPP between GL and HC at three overwintering stages. Red and blue letters show up-regulated and down-regulated proteins in the comparison groups, respectively. The same below.

Protein ID	Description	HC1 vs. GL1	HC2 vs. GL2	HC3 vs. GL3
A0A498JQC8	Glyceraldehyde-3-phosphate dehydrogenase	2.13	2.24	3.01
A0A540KTG1	Glyceraldehyde-3-phosphate dehydrogenase	1.99	-	2.38
A0A540M707	Probable 6-phosphogluconolactonase	1.78	1.87	-
A0A498I8M1	Malate dehydrogenase	1.61	-	1.94
A0A498JSH1	Phosphopyruvate hydratase	0.04	-	-
A0A540KR11	ATP citrate synthase	-	1.86	2.48
A0A540MIX1	Pyrophosphate--fructose 6-phosphate 1-phosphotransferase subunit beta	-	1.69	1.72
Q9ZPB7	Aldehyde dehydrogenase family 7 member A1	-	1.62	-
A0A540NAK7	Pyruvate kinase	-	0.63	-
A0A498ITD4	Glucose-6-phosphate dehydrogenase (NADP(+))	-	0.60	-
A0A1B1UZZ5	Phosphoenolpyruvate carboxykinase	-	0.60	-
A0A498HRZ9	Glucose-6-phosphate 1-dehydrogenase		0.60	
A0A498I194	Aldedh domain-containing protein	-	0.57	-
A0A498KGN9	Succinate dehydrogenase	-	0.57	-
A0A498HDN1	Succinate dehydrogenase	-	0.56	-
A0A540NBC6	Citrate synthase	-	0.55	-
A0A540LJY9	Glucose-6-phosphate isomerase	-	0.53	-
A0A498HE04	Citrate synthase	-	0.52	-
A0A498HN15	ATP citrate synthase	-	0.51	-
A0A498JRL5	Phosphoglycerate kinase	-	0.49	-
A0A498KQA5	ATP-dependent 6-phosphofructokinase	-	0.47	-
A0A540M0H0	ATP-dependent 6-phosphofructokinase	-	0.45	-
A0A498JVZ3	Fructose-bisphosphate aldolase	-	0.41	-
A0A498J4J5	Glucose-6-phosphate 1-dehydrogenase	-	0.39	-
A0A498KN40	Phosphopyruvate hydratase	-	-	3.03
A0A498IKR9	Pyruvate kinase	-	-	2.08
A0A540M4P5	Aldehyde dehydrogenase (NAD(+))	-	-	2.05
A0A498IVM7	Pyruvate decarboxylase	-	-	1.97
A0A498K862	Pyruvate kinase	-	-	1.71
A0A498JT62	Pyrophosphate--fructose 6-phosphate 1-phosphotransferase subunit alpha	-	-	1.64
A0A498HK68	Aldose 1-epimerase	-	-	0.54
A0A498JQT3	Pyruvate kinase	-	-	0.54
A0A498K2Y9	Dihydrolipoamide acetyltransferase component of pyruvate dehydrogenase complex	-	-	0.51

**Table 2 ijms-25-02964-t002:** List of fold-changes of DEPs in starch and sucrose metabolism.

Protein ID	Description	HC1 vs. GL1	HC2 vs. GL2	HC3 vs. GL3
A0A498HTE9	Ectonucleotide pyrophosphatase/phosphodiesterase family member 1/3	2.22	-	4.25
A0A498KGZ1	UTP--glucose-1-phosphate uridylyltransferase	0.63	-	0.57
A0A540L083	Glucose-1-phosphate adenylyltransferase	0.56	0.25	-
A0A498JZZ9	Trehalose 6-phosphate synthase	-	1.98	
A0A498INN2	β-glucosidase	-	1.85	
A0A498IVD8	Glycogen phosphorylase	-	1.58	
A0A498KD22	β-glucosidase	-	1.51	
A0A498JB71	Alpha-amylase	-	0.67	
A0A498ITV5	Glucan endo-1,3-beta-D-glucosidase	-	0.66	
A0A498J394	“Alpha-1,4 glucan phosphorylase”	-	0.66	
A0A498JBU0	Granule-bound starch synthase	-	0.59	
A0A540K631	UTP--glucose-1-phosphate uridylyltransferase	-	0.56	
A0A540LJY9	Glucose-6-phosphate isomerase	-	0.53	
A0A498K3I6	Alpha-amylase	-	0.51	
A0A498HCN5	“Alpha-1,4 glucan phosphorylase”	-	0.47	
A0A498IMZ2	β-glucosidase	-	0.45	
A0A540KIV9	Sucrose-phosphate synthase	-	0.39	
A0A498KQ57	Beta-fructofuranosidase	-	0.38	
A0A498JFR5	Beta-amylase	-		7.8
B2LUN5	“Starch synthase, chloroplastic/amyloplastic”	-		3.76
A0A498HVP0	Sucrose synthase	-		2.62
A0A498KKD2	Sucrose synthase	-		2.24
A0A498KD22	“Alpha-1,4 glucan phosphorylase”	-		2.24
A0A498IFG9	Amylomaltase	-		2.05
A0A498J394	“Alpha-1,4 glucan phosphorylase”	-		1.74
A0A498JXG5	Glucose-1-phosphate adenylyltransferase	-		1.72
A0A498HMG6	Glyco_transf_20 domain-containing protein	-		1.71
A0A498J684	Glucose-1-phosphate adenylyltransferase	-		1.67
A0A498HLR5	Endoglucanase	-		0.52
A0A498IIV3	β-glucosidase	-		0.46

**Table 3 ijms-25-02964-t003:** List of fold-changes of DEPs in oxidative phosphorylation.

Protein ID	Description	HC1 vs. GL1	HC2 vs. GL2	HC3 vs. GL3
A0A498HR73	NADH dehydrogenase (ubiquinone) 1 beta subcomplex subunit 10	2.65	-	-
A0A540L321	NADH dehydrogenase (ubiquinone) 1 beta subcomplex subunit 10	1.62	-	-
A0A498JWZ4	Inorganic diphosphatase	0.66	-	-
A0A498HS84	Cytochrome b-c1 complex subunit 6	0.65	-	-
A0A498KI10	“NADH dehydrogenase [ubiquinone] iron-sulfur protein 4, mitochondrial”	0.65	-	-
A0A498HIN9	NADH dehydrogenase (ubiquinone) 1 alpha subcomplex subunit 13	0.64	-	0.43
A0A540KHW1	Cytochrome c oxidase subunit 5b	0.61	-	-
A0A1C8YB78	“ATP synthase subunit b, chloroplastic”	-	0.66	-
A0A0U2PCG6	NADH dehydrogenase subunit 7	-	0.65	-
A0A540K7R9	NADH dehydrogenase (ubiquinone) Fe-S protein 8	-	0.65	-
A0A0U2N8T4	ATP synthase subunit alpha	-	0.63	-
A0A540LIW9	L51_S25_CI-B8 domain-containing protein	-	0.63	-
A0A498IGB9	CHCH domain-containing protein	-	0.61	-
A0A498K1S4	NADH dehydrogenase (ubiquinone) Fe-S protein 8	-	0.58	-
A0A498KGN9	“Succinate dehydrogenase [ubiquinone] flavoprotein subunit, mitochondrial”	-	0.57	-
A0A540LA69	Ubiquinol-cytochrome c reductase subunit 9	-	0.57	-
A0A498HDN1	“Succinate dehydrogenase [ubiquinone] iron-sulfur subunit, mitochondrial”	-	0.56	-
A0A540KIA4	Cytochrome b-c1 complex subunit 7	-	0.53	-
A0A540M5Y1	Acyl carrier protein	-	0.53	-
A0A498IPL8	Complex I-B22	-	0.51	-
A0A498HTR2	“NADH dehydrogenase [ubiquinone] flavoprotein 1, mitochondrial”	-	0.49	-
A0A498HI08	Cytochrome c oxidase subunit 5b	-	0.45	-
A0A498HA14	F-type H+-transporting ATPase subunit epsilon	-	0.38	-
A0A498HSD4	“NADH dehydrogenase [ubiquinone] iron-sulfur protein 4, mitochondrial”	-	0.36	-
A0A498KJ40	F-type H+-transporting ATPase subunit O	-	0.20	-
A0A498J104	Plasma membrane ATPase	-	-	1.85
A0A498IJL8	H(+)-exporting diphosphatase	-	-	1.58
A0A498ICJ5	V-type proton ATPase subunit G	-	-	0.65
A0A498ILV3	V-tcype proton ATPase subunit F	-	-	0.65
A0A498JAF9	Cytochrome c oxidase subunit 6b	-	-	0.65
A0A540MXE2	F-type H+-transporting ATPase subunit O	-	-	0.62
A0A498I904	NADH dehydrogenase (ubiquinone) 1 alpha subcomplex subunit 13	-	-	0.61
A0A498KNH1	“ATP synthase subunit d, mitochondrial”	-	-	0.56
A0A498KLQ7	Acyl carrier protein	-	-	0.46
A0A498K7I9	NADH dehydrogenase [ubiquinone] 1 alpha subcomplex subunit 1	-	-	0.44
A0A540LYH9	Acyl carrier protein	-	-	0.41
A0A540L2D7	Acyl carrier protein	-	-	0.38

**Table 4 ijms-25-02964-t004:** List of fold-changes of DEPs in phenylpropanoid biosynthesis and flavonoid biosynthesis.

Protein ID	Description	HC1 vs. GL1	HC2 vs. GL2	HC3 vs. GL3
A0A498KCP1	Peroxidase	0.46	0.37	-
A0A498HEU3	Shikimate O-hydroxycinnamoyltransferase	0.46	-	-
A0A498I2P0	Peroxidase	0.43	0.50	-
A0A540MB71	5-O-(4-coumaroyl)-D-quinate 3′-monooxygenase	-	1.64	-
A0A498KKN7	Peroxidase	-	1.62	2.24
H9U3A3	Cinnamate-4-hydroxylase	-	0.64	-
A0A498IIU4	Peroxidase	0.62	0.59	-
A0A498JXD0	Caffeoyl-CoA O-methyltransferase	-	0.38	-
A0A498KLW3	Peroxidase	-	-	2.10
A0A540NF19	Coniferyl-alcohol glucosyltransferase	-	-	1.95
C5IGQ5	Flavonoid 3′ hydroxylase IIb	-	-	0.62
A0A498J555	Fe2OG dioxygenase domain-containing protein	-	-	0.54
A0A498I8Y9	Caffeoyl-CoA O-methyltransferase	-	-	0.52

## Data Availability

All data supporting the findings of this study are available within the paper and within its Appendix A published online.

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
