# Peer review of "DIA-Based Quantitative Proteomics in the Flower Buds of Two Malus sieversii (Ledeb.) M. Roem Subtypes at Different Overwintering Stages"

_ijms, 2024, doi:10.3390/ijms25052964_

Round 1

Reviewer 1 Report

Comments and Suggestions for Authors

There are few english mistakes or typos to review but it is more readable

The conclusion should be ameliorated adding a conclusion also about the three overwintering bud stages.

Also the introduction can be improved adding some information of the state of the art in overwintering period studies

Fig. 2 represents a subclusterization and not exactly a hierachical clustering

Line 157:how is predicted the protein domain? and add ref

line 160: it has not a significant interest that metabolic process and cellular one are the major BP terms, it can be deepen

line 197: ref about WGCNA (also in M&M section)

Fig 4: must be ameliorated, in this form legend overlaps graphs and increase resolution

In the discussion (line 312-318): why authors inferred to copper stress? cold and copper stress differ also in ion flux. itshould be explain or better find other speculation

An experiment or analysis about phosphorylation it will be appreciated as authors note that oxidative phosphorylatio  takes place.

Comments on the Quality of English Language

There are few english mistakes or typos to review but it is more readable

Reviewer 2 Report

Comments and Suggestions for Authors

1- add conclusion at the end (Abstract)

2- Add objectives or goal of the study (Introduction)

3- Explains the process through which these proteins were identified (phases of dormancy) results

Time series analysis is missed in the manuscript (results)

Western blotting is not carried out of the protein samples (results)

add complete name of instruments used along with company and country name, please mention the name of database, where proteins taken, please mention the name of database, where proteins taken, Statistical analysis in not included in the methodology section (Materials and methods)

4- Add updated references in manuscript and must be within 5 years, old references must be removed

Comments on the Quality of English Language

English can be improved of the manuscript.
